# OpenReview forum: "Don't Walk the Line: Boundary Guidance for Filtered Generation"
_ICML.cc/2026/Conference — ICML 2026 regular_

### Official Review · Reviewer_y9rt · 2026-03-10

**Soundness:** 2
**Presentation:** 4
**Significance:** 2
**Originality:** 3
**Overall Recommendation:** 5
**Confidence:** 4

**Summary:**

This paper proposes Boundary Guidance: a term measuring the closeness between the (unsafe) score $t$ of the output of a generative model and the decision boundary $\tau$ of a safety classifier is included additively into the reward function during reinforcement fine-tuning. The authors claim that the inclusion of this term can lead to Pareto-improvement within the harmfulness-helpfulness space for LLMs.

**Compliance With Llm Reviewing Policy:**

Affirmed.

**Final Justification:**

The rebuttal addressed my main concerns.

**Key Questions For Authors:**

I would be happy to raise the Soundness and Significance scores if the authors can address the following:
1. Can the authors show the empirical distributions of $t$ values of samples produced by 1) base models and 2) models fine-tuned to reduce the probability of being filtered, to support the claim that motivated this paper?
2. Can the authors include an experiment that compares $R = u + \lambda m(t)$ and $R = u$?

Additionally, it would be beneficial if the authors can include:

3. An experiment that varies the strength of the proposed term (e.g., through $\lambda$).
4. Categorical rather than numerical evaluation results, as it is unclear to me how much better (if any) an average helpfulness of $2.26$ is more than $2.13$.

**Limitations:**

Yes, the authors adequately discussed the limitations and potential negative societal impact of their work.

**Strengths And Weaknesses:**

Strengths:
1. Soundness: This paper includes utility-theoretic analysis of the proposed term and extensive experiments and ablation studies.
2. Presentation: This paper is well-written and easy to follow. The proposed method is sufficiently motivated.
3. Significance: This paper addresses a potentially important and relevant problem to the safety and steerability of generative models.
4. Originality: This paper applies a decision-theoretic concept to LLM fine-tuning.

Weaknesses:
1. Soundness: The assumption regarding the score $t$ is too strong. The authors addressed this as a limitation.
2. Soundness: The authors did not discuss the possibility that output close to the decision boundaries (especially of weaker classifiers) can be particularly helpful. By uniformly avoiding the margin, helpfulness may be systematically negatively impacted in high-stakes applications.
3. Soundness: The numeric evaluations of the proposed method, despite their extensiveness, are insufficient to justify the claims:
  * The only metrics reported in this paper are harmfulness and helpfulness from LLMs-as-judges based on prompts unrelated to the actual reward and safety models used in fine-tuning. Harmfulness and helpfulness are related to but distinct from the score $t$ and utility $u$ discussed in the methodology section. As a result, many claims lack direct experimental evidence.
  * The authors did not compare $R = u + \lambda m(t)$ and $R = u$. Table 1 may not be a fair comparison as the reinforcement fine-tuning of the base models (if any) might have used a very different $u$ function than `Skywork-Reward-V2-Llama-3.1-8B-40M`.
  * I understand the authors could have chosen the then-latest models when the experiments were first conducted, but I fear the results might not be reproducible with the current models.
4. Significance: While the idea of avoiding the margin is definitely relevant, the way the proposed method operationalizes this idea seems straightforward from the motivation and lacks formal analysis. Furthermore, only the most basic form of the proposed term ($|t-0.5|$) is analyzed and evidenced by experiments.

---

> ### Author Rebuttal · Authors · 2026-03-30
>
> Thanks a lot for the valuable feedback!
> ## Weaknesses
> 1. (see also DbLJ) Thanks for pointing out that as written, the condition sounds strong. We will clarify that the result holds for the **expected** utility of an outcome under the base policy conditional on the score (i.e., $E[u | t]$) is monotonic. If the training objective only depends on the score (such as our guard_only reward, which performs the best), then it is without loss for the expected reward we are considering to only operate on $E[u | t]$. Under this view, the assumption of Theorem 1 operates on the average utility of the base model conditional on a score t, which we believe is credible (e.g, we demand that the societal utility of refusal increases with unsafety-"unsafety is bad on average"-and the societal utility of generation decreases with unsafety- (”overblocking is bad on average”). We will clarify how our condition can be viewed via a conditional expectation
>
> 2. With the interpretation given to Weakness 1, we struggle to come up with an example of why the expected utility should've a local maximum at the decision boundary
>
> 3.1. We will clarify how our experiments test a “robust” version of our theoretical claims: The utility function is not known to us and likely hard to formulate even for deployers. We therefore consider **Pareto improvements** on helpfulness and harmlessness, which would mean utility improvements for any weighting of these two (at times competing) results. Independently, we believe that following [1]’s prompts (and others’, see section 4.2) based on stronger commercial LLM judges, we hope for meaningful comparisons and higher construct validity of our evaluation
>
> 3.2. Thanks, see our answer to Question 2 below!
>
> 3.3. Yes, results change each model generation and affects resubmitted papers in particular
>
> 4. We might miss something in your first point: Being straightforward from the motivation should only improve the impact of a method. Concerning your second point: We agree. Choosing different boundary guidance functions (which one might think of as being akin to temperature scaling for the classifier) could only increase the performance of the best boundary-guided finetuning. By not optimising (due to computational limitations), we stacked the deck against us.
>
> ## Questions
> 1. Table A below shows the percentage of t outside the interval of 0.1 to 0.9, which roughly shows that our finetuning is able to maximize its given reward. Comparing base to the fine-tuned version of the model (reward_and_guard) shows that we constantly push t to more extreme values on every dataset. Note that naturally this improvement is most visible in the jailbreak evaluation dataset (Test JB) because here more examples lie closer to the classifier's decision boundary. We include this table in the revision
>
> TABLE A: Percentage of t outside the interval of 0.1 to 0.9:
>
> | Model | Experiment Type | Test JB | Test Harm | Test Benign |
> |------------|-----------------|-------------|---------|--------------|
> | Qwen25-0.5b | base |  0.80 | 0.89 | 0.99 |
> |  | reward_and_guard |  0.97 | 0.99 | 1.00 |
> | Qwen25-7b | base |  0.87 | 0.87 | 1.00 |
> | | reward_and_guard |  0.97 | 0.98 | 1.00 |
> | Qwen25-14b | base |  0.92 | 0.92 | 1.00 |
> |  | reward_and_guard |  1.00 | 0.98 | 1.00 |
> | Gemma-2-9b | base |  0.93 | 0.98 | 0.99 |
> |  | reward_and_guard |  0.97 | 1.00 | 1.00 |
>
> 2. We also finetuned the proposed baseline with $r=u$, and report results compared to the base model in Table B.
>
>    TABLE B: Results for finetuning against the Skyworks model
>
> | Metric | FT | Base | Δ | FT | Base | Δ | FT | Base | Δ | FT | Base | Δ |
> |---|---|---|---|---|---|---|---|---|---|---|---|---|
> | | Qwen2.5-0.5B | | | Qwen2.5-7B | | | Gemma-2-9B-it | | | Qwen2.5-14B | | |
> | Helpful | 2.38 | 2.13 | **+0.25\*\*\*** | 2.76 | 2.72 | +0.04 | 2.65 | 2.63 | +0.02 | 2.79 | 2.77 | +0.03 |
> | Harmful | 0.15 | 0.27 | **−0.12\*\*\*** | 0.03 | 0.19 | **−0.17\*\*\*** | 0.11 | 0.07 | **+0.04\*** | 0.05 | 0.14 | **−0.09\*\*\*** |
>
> Compared to the $u+|t - 0.5|$ (eqn. (1)) reward, we observe ambiguous results: Finetuning against $u$ benefits the smallest (0.5B) model beyond eqn. (1), suggesting that this model is not capable to generalize for the boundary guidance reward. For the second-biggest (9B) model, we find decreased harmfulness, for the largest (14B) model decreased helpfulness. No other differences can be detected.
>
> The comparison to our most performant finetuning reward, $R’$, is unambiguous, however, for all >0.5B models: $R’$ either matches or outperforms $u$ on all models and metrics.
>
> 3. Varying the fine-tuning reward would improve the performance of the best boundary-guided fine-tuning. We are happy to include it for the camera-ready version (adding it will only improve the evidence presented here).
> 4. The numerical evaluation is based on a categorical rubric, see Appendix B.
>
> [1] Yuan et al. "From hard refusals to safe-completions: Toward output-centric safety training." 2025.

---

> > ### Author Rebuttal · Reviewer_y9rt · 2026-04-04
> >
> > Thanks for your rebuttal. My concerns have been adequately addressed. I will raise my overall recommendation. Please include the items discussed in the camera-ready version. Please also consider improving the presentation of the figures and the tables.
> >
> > Regarding my 1st point for Significance: There is more than one way to operationalize the motivation of boundary avoidance. Explicitly modeling it as a term in the reward seems a bit arbitrary and heuristic and requires more assumptions than otherwise necessary. Without the assumption of Theorem 1, one might interpret strong boundary avoidance as a potential force to decrease the diversity of generated responses, which may lead to missed local maxima and mode collapse. I acknowledge the significance of the utility-theoretic framework, but it seems to me a somewhat oversimplification of the reality. That being said, I enjoyed reading this paper and recommend its acceptance.

---

> > > ### Author Response · Authors · 2026-04-04
> > >
> > > Thank you for your positive and concrete review, which improved the paper; we are looking forward to your increased score!
> > >
> > > We will include the discussed items in the camera-ready version and work over figures and tables. We will also make an effort to discuss the assumptions for our theory clearly.

---

### Official Review · Reviewer_p29H · 2026-03-11

**Soundness:** 3
**Presentation:** 3
**Significance:** 3
**Originality:** 3
**Overall Recommendation:** 5
**Confidence:** 3

**Summary:**

The authors tackle a very common setting (especially in commercial use of LLM), where an LLM is paired with a downstream classifier that is tasked to filter out content based on, e.g., harmfulness and unhelpfulness. Standard alignment techniques can push the LLM to generate outputs that sit very close to the downstream classifier's decision boundary, which can lead to sub-optimal performance and false positives and negatives scenarios. The authors propose to include an explicit reward in the RL fine-tuning objective that pushes the LLM output away from the margin, by combining the GRPO loss with an utility reward based on the distance of the output to the boundary. The experiments show (often statistically significant) improvements in helpfulness and reduction in harmfulness.

**Compliance With Llm Reviewing Policy:**

Affirmed.

**Final Justification:**

Between the time from having read and reviewed the paper I have familiarized myself more with this literature and, considering the authors rebuttal, I have kept my score consistent (as I believe this paper would be a good contribution) but increased my confidence.

**Key Questions For Authors:**

- Can you elaborate on the theoretical or empirical threshold for classifier quality required for your approach to be effective (see comment above)?
- How sensitive is your approach on the RL optimization algorithm? Have you tried approaches with PPO or DPO?
- [Minor] How do you anticipate the approach to work for complex reasoning task such as code, where one might wanna reduce the chance of harmful code but pushing the model away from a safety margin might disrupt a necessary logical chain?

**Limitations:**

Yes

**Strengths And Weaknesses:**

I wanna preface the below by saying that I'm not an expert in alignment and safety research.

Strenghts:
- The premise is straightforward but highly practical. Penalizing outputs that fall in the ambiguous margin of a safety classifier is a good approach to a real-world deployment problem
- The theoretical justification is relative simple, but formalizes a series of desirable behavior the utility function would have, which strengthen the mathematical foundation of the proposed reward function
- Experiment results often show statistical significant improvement in helpfulness and reduction in harmfulness

Weakness:
- The empirical validation just considers the GRPO + Lora fine-tuning, leaving open the questions of whether other finetuning approaches (e.g., PPO) would also benefit from the proposed reward
- The model seems to be highly dependent on a very good "filter" model, as using a weaker classifier seem to make the result getting worse. This might be complicated where the "filtering" needs to be exercised for less clear-cut situations, e.g., jailbreaking.
- [Minor] The evaluation focuses heavily on standard instruction following, but pushing a model away from the decision boundary might negatively impact downstream complex reasoning task where nuance might be required

---

> ### Author Rebuttal · Authors · 2026-03-30
>
> Thanks a lot for your engagement and the valuable feedback! We address the weaknesses and questions next:
>
> ## Weaknesses
> 1. Due to computational limits we did not run full-parameter fine-tuning, but as this is another model, it would only improve the performance of our best model compared to the base model. \
> We are happy to provide a PPO fine-tuning run for our main loss function and report results in an appendix for the camera-ready version. We have reason to believe that DPO is strictly less performant than GRPO/PPO, as it would binarize a rich, real-valued reward, increasing sample complexity of the method without apparent upside.
>
> 2. We agree that our method relies on the quality of the filter model. We will add to the Limitations section that our fine-tuning is limited by the quality of available filters.
>
> 3. This is correct and a challenge for better evaluations. We chose helpfulness and harmlessness based on [1] to make the evaluation simple and comparable to an existing paper.
>
> ## Questions
> 1. The interesting answer to this question is that if the monotonicity condition in Proposition 1 are given for a classifier of choice—entirely independently of noise model, then, by Proposition 1, finetuning with boundary guidance, should improve societal utility. (This does not mean that formulating the utility function is easy–it is unknown to us, and in general, hard to formulate. Therefore, our experiments test a robust version of the hypothesis of improving utility: If we think of utility of the model as being some weighting of helpfulness and the negative of harmlessness, then a Pareto improvement in both means a utility improvement for *any* weights.)
> 2. See Weakness 1.
> 3. The models we consider in this paper are quite small, because we chose to balance the breadth of experiments we run and the computational resources at our disposal. We believe that evaluating on code generation is a great area for future research, in particular in light of recent announcements of Anthropic to use classifiers for Claude Code permissioning [2].
>
> ## References
> [1] Yuan, Yuan, et al. "From hard refusals to safe-completions: Toward output-centric safety training." arXiv preprint arXiv:2508.09224 (2025). \
> [2] https://www.anthropic.com/engineering/claude-code-auto-mode

---

> > ### Author Rebuttal · Reviewer_p29H · 2026-04-03
> >
> > Thanks for the rebuttal! I consider my questions mostly addressed by the authors, and I have increased the confidence in my score accordingly. I have a couple of quick comments:
> >
> > - While I appreciate your theoretically justification on weakness 1, it leaves open the question on what the minimum capacity of the classifier should be, which I think should be acknowledged as open question
> >
> > - Please include the results of a PPO run, although your intuition on how the results would turn out makes sense to me

---

> > > ### Author Response · Authors · 2026-04-04
> > >
> > > Thank you for your careful response. We will acknowledge the capacity question in our conclusion as an open problem, and add a PPO run.

---

### Official Review · Reviewer_DbLJ · 2026-03-12

**Soundness:** 2
**Presentation:** 3
**Significance:** 2
**Originality:** 2
**Overall Recommendation:** 4
**Confidence:** 3

**Summary:**

This paper proposes Boundary Guidance, a reinforcement learning fine-tuning method that encourages a generator to produce outputs that are clearly on one side of a safety classifier’s decision boundary. The key idea is to add a reward term proportional to the distance from the classifier threshold, with the goal of reducing both over-refusals and unsafe outputs.

**Compliance With Llm Reviewing Policy:**

Affirmed.

**Final Justification:**

The authors have reframed some of the claims. I have asked them to remove the theorem statement.

**Key Questions For Authors:**

See above

**Limitations:**

yes

**Strengths And Weaknesses:**

Strengths

1. The paper addresses a relevant and practical problem: the interaction between generative models and safety filters in compound systems.

2. The proposed reward shaping mechanism is simple and easy to implement.

3. Experiments are conducted across multiple model sizes and consistently show reductions in harmful outputs.

Weaknesses

1. The main issue of the paper is the theoretical framing. Several results in Section 3 appear largely tautological and derive properties that are already encoded in the assumptions. Theorem 1 assumes that the utilities for rejecting and passing are monotone in opposite directions and intersect at the threshold $\tau$. Under these assumptions it is immediate that the maximum of the two utilities has its minimum at $\tau$. While correct, presenting this as a theorem does not provide additional insight beyond the assumptions themselves.
Similarly, Proposition 1 derives monotonicity of the expected rejection utility by imposing monotonicity assumptions directly on the underlying utility functions $u(\hat z, z, t)$ and on $p(t)=P(z=1|t)$. The conclusion therefore largely follows from the structure imposed on the components.

2. The analysis relies on the assumption that the utility of accepting or rejecting an output depends only on the classifier score $t(y)$ and not on the output $y$ itself. This effectively treats the classifier score as a sufficient statistic for utility. In realistic LLM safety settings this assumption is quite strong: outputs with similar classifier scores can differ significantly in semantic content and downstream utility. As a result, the monotonicity arguments and the boundary-guidance intuition may not hold in practice.

3. The empirical improvements reported in Table 1 are relatively small, and helpfulness decreases slightly for the largest model. Given the simplicity of the proposed reward shaping and the limited theoretical support, the current evidence does not yet convincingly establish the method as a broadly applicable approach.

---

> ### Author Rebuttal · Authors · 2026-03-30
>
> Thanks a lot for your engagement and the valuable feedback! We address the weaknesses and questions:
>
> ## Weaknesses
> ### 1
> We agree that both Theorem 1 and Proposition 1 are intuitive. We view the main contribution of these two results in their (decision-theoretic) *formalization* of when Boundary Guidance is expected to work, not the theoretical hardness of the results. We would like to clarify, however, that the results are nontrivial. Proposition 1 is a set of sufficient conditions for the monotonicity condition of Theorem 1. The monotonicity assumption in Theorem 1 is about an implicit object: the utility of an action, which depends on the distribution of utility and safety score. Proposition 1 is a sufficient condition for this implicit object to satisfy monotonicity with respect to score. While hence Proposition 1 is less direct, Theorem 1 is labelled as a Theorem, as it is the main conceptual insight of the paper, not because of the hardness to prove it—it is a one-liner.
>
> ### 2
> We are grateful for you pointing out that as written, the condition sounds very strong. We will make clearer that the result holds for the **expected** utility of an outcome under the base policy conditional on the score (i.e., $E[u | t]$) is monotonic. If the training objective only depends on the score (such as our guard_only reward, which performs the best), then it is without loss for the expected reward we are considering to only operate on $E[u | t]$. Under this view, the assumption of Theorem 1 operates on the average utility of the base model conditional on a score t, which we believe is credible (for example, we demand that the societal utility of refusal increases with unsafety ("unsafety is bad on average") and the societal utility of generation decreases with unsafety (”overblocking is bad on average”). We will make clearer how our condition can be viewed via a conditional expectation.
>
> ### 3
> Concerning the size of improvements: Our improvements are comparable to results in related papers. For example, consider Figure 4(a) right and (b) right in [1], which is the paper from which we sourced our judge prompts. The improvements on benign and dual-use prompts conditional on the output being safe are 0.15, 0.23, 0.08 and 0.25, respectively. (4(a) left and 4(b) left are much smaller, but not comparable, as binary vectors. The malicious prompts have effects of one order of magnitude larger than this, but are not directly comparable as these are conditional on safe outputs to malicious prompts.)
>
> We also highlight that a simple method like ours is good for potential impact, and not a liability.
>
> Given that the concerns about our theory seem to be a main driver of your review, we invite you to raise your score to (weak) accept based on these two clarifications.
>
> ## References
> [1]: Yuan, Yuan, et al. "From hard refusals to safe-completions: Toward output-centric safety training." arXiv preprint arXiv:2508.09224 (2025).

---

> > ### Author Rebuttal · Reviewer_DbLJ · 2026-04-03
> >
> > I have updated my score to reflect the answers. I would ask the author to remove the main statement from “theorem”.

---

> > > ### Author Response · Authors · 2026-04-04
> > >
> > > Thank you for the update, and we are happy to do this! Could you clarify what you mean by "removing the main statement"? Would you like us to remove the label Theorem? We are happy to call it a Proposition, for example.

---

### Decision · Program_Chairs · 2026-04-30

**Decision:**

Accept (regular)

**Comment:**

**Summary:** This paper proposes Boundary Guidance, a reinforcement learning fine-tuning method that steers LLM generation away from a safety classifier's decision boundary. The approach adds a reward term proportional to the distance from the classifier threshold during GRPO fine-tuning. Experiments across multiple model scales show improvements in both helpfulness and harmfulness metrics.

**Strengths:** Addresses a practical and well-motivated problem in compound LLM systems. The method is simple and easy to implement. Extensive ablations across model scales and reward designs. Well-written paper with clear presentation.

**Weaknesses:** The theoretical results, while formalizing the intuition, rely on assumptions that largely encode the conclusions. The method depends on classifier quality, and the minimum capacity required remains an open question. Empirical improvements, though often statistically significant, are modest in magnitude.

**Recommendation:** Accept.